# Make an Omelette with Breaking Eggs: Zero-Shot Learning for Novel Attribute Synthesis

**Yu-Hsuan Li**\*
National Chiao Tung University
evali890227@gmail.com

**Tzu-Yin Chao**\*
National Chiao Tung University
chaotzuyin@nctu.edu.tw

**Ching-Chun Huang**
National Chiao Tung University
chingchun@cs.nctu.edu.tw

**Pin-Yu Chen**
IBM Research
pin-yu.chen@ibm.com

**Wei-Chen Chiu**
National Chiao Tung University
walon@cs.nctu.edu.tw

## Abstract

Most of the existing algorithms for zero-shot classification problems typically rely on the attribute-based semantic relations among categories to realize the classification of novel categories without observing any of their instances. However, training the zero-shot classification models still requires attribute labeling for each class (or even instance) in the training dataset, which is also expensive. To this end, in this paper, we bring up a new problem scenario: "*Can we derive zero-shot learning for novel attribute detectors/classifiers and use them to automatically annotate the dataset for labeling efficiency?*". Basically, given only a small set of detectors that are learned to recognize some manually annotated attributes (i.e., the seen attributes), we aim to synthesize the detectors of novel attributes in a zero-shot learning manner. Our proposed method, **Z**ero-**S**hot **L**earning for **A**ttributes (ZSLA), which is the first of its kind to the best of our knowledge, tackles this new research problem by applying the set operations to first decompose the seen attributes into their basic attributes and then recombine these basic attributes into the novel ones. Extensive experiments are conducted to verify the capacity of our synthesized detectors for accurately capturing the semantics of the novel attributes and show their superior performance in terms of detection and localization compared to other baseline approaches. Moreover, we demonstrate the application of automatic annotation using our synthesized detectors on Caltech-UCSD Birds-200-2011 dataset. Various generalized zero-shot classification algorithms trained upon the dataset re-annotated by ZSLA shows comparable performance with those trained with the manual ground-truth annotations. Please refer to our project page for source code: https://yuhsuanli.github.io/ZSLA/

## 1 Introduction

Zero-shot learning (ZSL) algorithms for classification aim to recognize novel categories without observing any of their instances during model training; thus, the cost of collecting training samples for the novel categories can be eliminated. Typically, the core challenge behind zero-shot classification lies in associating novel categories with the seen ones during training. Various existing approaches leverage different auxiliary semantic information to construct such associations across categories, thus being able to generalize the learned models for classifying novel categories [17, 1, 32, 16, 3, 37] or synthesize the training samples for each novel category [39, 40, 33, 43, 26]. Among different types of auxiliary semantic information adopted for ZSL, defining a group of attributes shared among

---

\*The authors contributed equally to this work.
36th Conference on Neural Information Processing Systems (NeurIPS 2022).

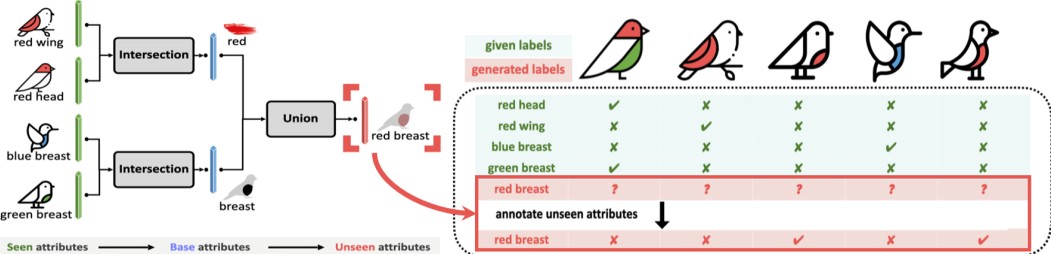

Figure 1: Given a set of trained/seen attribute detectors (e.g. "red wing", "red head", "blue breast", and "green breast"), our ZSLA can synthesize a novel detector for the unseen attribute (e.g. "red breast") by the following process: (1) applying the intersection operation on the subsets {"red wing", "red head"} and {"blue breast", "green breast"} respectively to extract the common semantics of each subsets, i.e. "red" and "breast", as the ***base attributes***; (2) combining the base attributes via the union operation to realize the novel/unseen attribute detector, i.e. "red breast". The novel attribute detectors can later be applied to annotate the dataset.

categories becomes one of the most popular choices, where each category is described by multiple attributes (i.e., multi-labeled by the attributes), and the attribute-based representations are discriminative across categories. However, it comes with the expensive cost of manually annotating the samples in the dataset their attribute labels at a much granular level. For example, CUB dataset [35], one of the most widely-used benchmarks for learning zero-shot classification, is built by spending a great deal of time and effort to label 312 attributes for 11788 images.

As motivated by the issue of annotation efficiency on attribute labels, this paper aims to ***develop ZSL on known attributes to annotate novel attributes for a dataset automatically*** . That is, analogous to the zero-shot classification scenario, we now advance to annotate novel attributes for a dataset via utilizing the knowledge from a few types of seen/given manual attributes, as illustrated in Figure 1. Specifically, we take the well-known CUB dataset [35] as our main test-bed and have a deep investigation on its attributes. We discover that, many attributes in CUB dataset (e.g. "red head" or "blue belly") follow the form of combinations over ***base attributes*** (e.g. "red", "blue", "head" and "belly" respectively). Building upon such observation, given a defined set of attributes in the form of the ones used in the CUB dataset and labels of a few ***seen attributes*** (where the number is far less than that of overall defined attributes), we propose **Z**ero-**S**hot **L**earning for **A**ttributes (ZSLA), a method of training the ***seen attribute detectors*** and then tackle the ZSL problem to synthesize unseen attribute detectors via a ***decompose-and-reassemble*** manner. In detail, the seen attribute detectors are firstly decomposed into base attribute representations, in which they are further reassembled with novel combinations into novel attribute detectors, as illustrated in Figure 1. Here, both the decomposition and reassembly steps are achieved via set operations (i.e., the interaction and union operators, respectively). Together with the seen ones, the novel attribute detectors can be utilized to annotate the attribute labels for the dataset automatically.

To demonstrate the efficiency of ZSLA, we synthesize 207 novel attribute detectors by leveraging only 32 seen ones from the CUB dataset. These novel attribute detectors are shown to be effective in capturing their corresponding semantic information and benefit both the attribute detection and localization for the samples in CUB dataset. Besides, we also synthesize $\alpha$-CLEVR dataset by [13] for conducting the controlled experiments to further discuss the influence of noisy seen attribute labels. The results show that ZSLA can provide more robust annotations than the other baseline methods under the noisy scenario. Below, we highlight the contributions of this paper:

- To the best of our knowledge, we are the first to propose ZSL for attributes to automatically annotate attribute labels for the zero-shot classification datasets.

- We propose a novel decompose-and-reassemble approach to single out the base attribute representations by applying intersection on the seen attribute detectors and synthesize the unseen ones by having the union operation over the base attributes representations.

- In Section 4.2, we show on the CUB dataset that, given only 32 attributes with manual annotations, ZSLA can synthesize novel attribute detectors to provide high-quality annotations for the dataset. By using the auto-annotated attributes, generalized zero-shot classification algorithms can also achieve comparable or even better performance than that using 312 manually-annotated attributes.

## 2 Related Works

Zero-shot learning (ZSL) was originally proposed to tackle the specific classification problem, where the model is expected to be capable of classifying the samples belonging to the novel categories which are not seen previously during training. The problem setup has been extended to other applications such as detection [5, 31, 9] and segmentation [6, 45]. Here we provide a brief review of the works of zero-shot classification [17, 32, 1, 2, 16, 41, 39, 40, 33, 43, 26]. Without loss of generality, the ZSL approaches rely on using the auxiliary information (such as attributes, word embeddings, or text descriptions) as the basis for describing the categories and building the semantic relation among seen and unseen categories; and the existing methods can be roughly categorized into two groups: the embedding-based methods [1, 2, 16, 33, 41] and generative methods [39, 40, 33, 43, 26]. The embedding-based methods basically aim to learn a latent space that connects between the feature representations of training samples and the embeddings of their corresponding auxiliary information (e.g., the visual features and the embeddings of attribute labels for the training images in the CUB dataset), such that the test samples can be classified as the novel categories once their feature representations are close to the embeddings of novel categories (which are defined upon auxiliary information without requiring any additional training samples). The generative methods instead utilize the deep-generative models (e.g., generative adversarial networks [11], variational autoencoder [15], or their hybrids/variants) for learning to synthesize the samples or features of the unseen categories based on their auxiliary semantic information. Though saving the effort of collecting the training samples to recognize novel categories via ZSL techniques, manually annotating the auxiliary semantic information for the samples in the zero-shot training dataset is still quite expensive and time-consuming. In turn, our proposed task of zero-shot learning for novel attributes helps to reduce such costs for the scenario of zero-shot classification where the auxiliary information is defined on attributes.

Aside from the typical zero-shot classification problem, *compositional zero-shot learning* (CZSL) and *blind source separation* (BSS) are tasks that our work is conceptually related to. CZSL [23, 25, 4, 21, 12, 24], also as known as *state-object compositionality* problem, aims to recognize the novel compositions (e.g. "ripe tomato") given the seen visual primitives of states (e.g. "ripe", "rotten") and objects (e.g. "apple", "tomato") in the training dataset. Our proposed problem scenario ZSLA is distinct from CZSL under several perspectives: (1) An image in our problem scenario would have multiple attributes while there usually exists only a single state-object composition for CZSL; (2) Our synthesized attribute detectors are able to provide labels of novel attributes (i.e. these novel attributes do not have any manually labeled samples in the training set) for the images thus leading to more detailed descriptions for all the categories, while CZSL typically aims to increase the number of categories (i.e. each novel composition is treated as a new fine-grained class). We provide the extensive discussions and experiments with respect to CZSL in Appendix D.3.

On the other hand, BSS[7, 36, 42, 8] aims to separate and distinguish the source of several signals from their mixture. Though our ZSLA framework composed of a decompose-and-reassemble procedure seems to be similar to blind source separation at first glance, there exists a significant distinction that differentiates our ZSLA from BSS in signal processing: Our intersection operation to perform decomposition on seen attributes is non-blind, in which it works by the guidance of logical and semantic constraints (i.e. two input attributes should have a common ground in one of the base attributes but not both); by contrast, there is no such constraint in the blind source separation (which instead typically adopts independent assumption or mutual information in its modeling).

## 3 ZSLA: Proposed Method

Given a zero-shot classification dataset $\{\mathbf{X}, \mathbf{Y}, \mathbf{A}^s\}$, each image $x \in \mathbf{X}$ has its class label $y \in \mathbf{Y}$ and the multi-attribute labels $\phi^s(x)$, where $\phi^s(x)$ is a binary vector with its each element denoting if $x$ has a certain attribute $a \in \mathbf{A}^s$. ZSLA starts with using $\{\mathbf{X}, \mathbf{A}^s\}$ to train the detectors $M^s$ for all the attributes in $\mathbf{A}^s$, which are treated as seen attributes, then it adopts the seen attribute detectors $M^s$ to synthesize the detectors $M^u$ for the unseen attributes $\mathbf{A}^u$ via a decompose-and-reassemble procedure, where $\mathbf{A}^s \cap \mathbf{A}^u = \emptyset$. For ease of understanding, we use the most popular zero-shot classification dataset, CUB [35], to illustrate how these steps are realized as follows.

## 3.1 Training Seen Attribute Detectors

Our attribute detectors are built on top of the image feature space produced by the image feature extractor $f$. Given an input image $x$ and its feature map $f(x) \in \mathbb{R}^{W \times H \times C}$ where each $C$-dimensional feature vector at position $(i, j)$ of $f(x)$, denoted as $f(x)[i, j]$, is the feature representation of the corresponding image patch on $x$, the attribute detectors $M^s \in \mathbb{R}^{C \times N^s}$ (in which $N^s$ denotes the number of attributes in $\mathbf{A}^s$) aim to give high response on the image patches containing the visual appearance related to the attributes in $\mathbf{A}^s$. Specifically, each column in $M^s$ is acting as the embedding of a certain attribute. We use $m_k^s$ to indicate the $k$-th column of $M^s$. The response of the corresponding $k$-th attribute in $\mathbf{A}^s$ with respect to the patch-wise feature vector $f(x)[i, j]$ is calculated by a specific form of their cosine similarity $\cos(|m_k^s|, f(x)[i, j])$, where $|m_k^s|$ denotes applying element-wise absolute-value operator on $m_k^s$. We have $|m_k^s|$ in our cosine similarity computation due to the reason that: Each dimension along channels of $f(x)$ is considered to capture a specific visual pattern. Our $|m_k^s|$ hence acts as to apply the weighted combination over these various visual patterns for representing the characteristics of the $k$-th attribute in $\mathbf{A}^s$, and the absolute-value operator over $m_k^s$ is to ensure the combination weights are non-negative.

Figure 2: Overview of Our ZSLA. **(1)** Training the seen attribute detectors: Seen attribute detectors, defined as the embeddings for each seen attribute, are built on top of the image features and their training is guided by two objectives: $\mathcal{L}_{bce}$ and $\mathcal{L}_{umc}$, where the former drives the trained detectors to perform binary classification for attributes on image patches (cf. Eq. 2) while the latter enforces the uni-modal constraint on the response map $\mathcal{R}^s(x)$ of patch-wise image features with respect to each attribute, in order to make it compact and concentrated (cf. Eq. 3). **(2)** Learning to synthesize novel/unseen attribute detectors via a decompose-and-reassemble procedure: Given the trained detectors of seen attributes, the intersection operation is firstly applied on them to extract base attributes, and then these base attributes are further combined by union operation to synthesize the novel/unseen attributes. The training of these operations is driven by the reconstruction loss $\mathcal{L}_{rec}$ (cf. Eq. 5) once the synthesized attribute coincides with any of the seen ones.

We denote $\mathcal{R}^s(x) \in \mathbb{R}^{W \times H \times N^s}$ as the response map which has included the cosine similarities of all the seen attributes $\mathbf{A}^s$ at each position on $f(x)$. Note that, as our feature extractor $f$ adopts the ReLU activation function in its last layer (similar to most image feature extractors based on the convolutional networks), the values in $f(x)$ become non-negative. Furthermore, as both $|m_k^s|$ and $f(x)[i, j]$ are non-negative vectors, all entries of $\mathcal{R}^s$ results to be within the range $[0, 1]$. Following the popular tricks for ZSL and deep learning pointed out in [34], where adopting scaled cosine similarity in logits computation is important to achieve better model training, we use the computation below to calibrate the value of elements in $\mathcal{R}^s(x)$:

$$\tilde{R}^s(x) = \gamma^2 \cdot (2 \cdot R^s(x) - 1) \tag{1}$$

where the calculation within brackets shifts and expands the values in $R^s(x)$ towards $[-1, 1]$ to match the typical value range of cosine similarity, and the hyperparameter $\gamma$ is set to 5 as suggested by [34]. Then, we perform the max-pooling operation on $\tilde{R}^s(x)$ and obtain the image-wise attribute response $\tilde{r}^s(x) \in \mathbb{R}^{N^s}$. Such logits over attributes thus are able to drive the model training (i.e. optimization over $M^s$ and $f$) via the error between the attribute detection results and the ground-truth attribute labels $\phi^s(x)$. The objective function $\mathcal{L}_{bce}$ to evaluate the error between the logits of attribute detection result $\tilde{r}^s(x)$ and the ground-truth attribute labels $\phi^s(x)$ is defined via the binary cross-entropy:

$$\mathcal{L}_{bce} = -\sum_k^{N^s} \phi_k^s(x) \cdot \log(\sigma(\tilde{r}_k^s(x))) + (1 - \phi_k^s(x)) \cdot \log(1 - \sigma(\tilde{r}_k^s(x))) \tag{2}$$

where $\phi_k^s(x)$ and $\tilde{r}_k^s(x)$ denote the $k$-th elements in $\phi^s(x)$ and $\tilde{r}^s(x)$ respectively, and $\sigma$ is the sigmoid function.

In addition to the $\mathcal{L}_{bce}$ loss, we introduce another objective function $\mathcal{L}_{umc}$ to place the **uni-modal constraint** [44] on the response map $\tilde{R}^s(x)$, which encourages the response map for a certain attribute (e.g. $\tilde{R}_k^s(x)$, the $k$-th channel of $\tilde{R}_k^s(x)$) to be uni-modal and concentrated. In other words, we regularize a detector to focus on a single location or a small region in the image $x$. We provide a further discussion of relaxing this assumption in Appendix A.4.

$$\mathcal{L}_{umc} = \sum_k^{N^s} \sum_{(i,j)} \sigma(\tilde{R}_k^s(x)[i,j]) \cdot \left( \left\| i - \breve{i}_k \right\|^2 + \left\| j - \breve{j}_k \right\|^2 \right), \tag{3}$$

where $\breve{i}_k, \breve{j}_k = \arg\max_{i,j} \tilde{R}_k^s(x)[i,j]$ and $\| \cdot \|$ denotes the Euclidean norm.

The overall objective to train the feature extractor $f$ and the seen attribute detectors $M^s$ is illustrated in the left portion of Figure 2 and summarized as: $\mathcal{L}_{bce} + \lambda \mathcal{L}_{umc}$, where the hyperparameter $\lambda$ controls the balance between losses and is set to $0.2$ in our experiments.

Moreover, we are aware that in CUB dataset the additional annotations indicating the ground-truth locations for the attributes which an image $x$ has are also available (e.g. we know where the attribute "brown wing" appears on an image of "gadwall"). Hence, in addition to max-pooling the response map $\mathcal{R}^s(x)$ to obtain the image-wise response $r^s(x)$ for attributes, we experiment another way to obtain $r^s(x)$: (1) If $\phi_k^s(x)$ is true, the $k$-th element in $r^s(x)$, i.e. $r_k^s(x)$, is assigned by $\mathcal{R}^s(x)[i,j]$ where the centre of the ground-truth location for the $k$-th attribute in $\mathbf{A}^s$ is located on the patch related to the position $(i, j)$ of $\mathcal{R}^s$; (2) If $\phi_k^s(x)$ is false, $r_k^s(x)$ is assigned by having the average pooling over the $k$-th channel of $\mathcal{R}^s(x)$. Appendix D.1 provides the analysis for the impact of using such additional annotations of attribute location on the performance of ZSLA.

## 3.2 Decompose-and-Reassemble for Synthesizing Novel Attribute Detectors

After obtaining the seen attribute detectors $M^s$, we now aim to perform the decompose-and-reassemble procedure (as shown in the right-half of Figure 2) for generating the detectors $M^u \in \mathbb{R}^{C \times N^u}$ of the novel attributes $\mathbf{A}^u$ (where $N^u$ is the number of attributes in $\mathbf{A}^u$) by leveraging $M^s$.

First, we observe that most of the attributes in CUB dataset (the most popular zero-shot classification dataset and also our test-bed in this work) follow the form of "*adjective + object part*", for instance: "black eye", "brown forehead", "red upper-tail", or "buff breast". Starting from such observation, we define two disjoint sets of **base attributes**, $\mathbf{B}^c$ and $\mathbf{B}^p$, representing the *adjectives* and *object parts* used in the seen attributes, respectively (e.g. "blue", "yellow", "solid", and "perching-like" for $\mathbf{B}^c$; "leg", "beak", "belly", and "throat" for $\mathbf{B}^p$). Please note that the concepts behind adjectives $\mathbf{B}^c$ in CUB dataset include not only color but also texture, shape, and others. Formally, given an attribute $a$, we use $\beta^c(a)$ and $\beta^p(a)$ to denote its corresponding base attributes on the adjective and object part, respectively (i.e. $\beta^c(a) \in \mathbf{B}^c$ and $\beta^p(a) \in \mathbf{B}^p$), where $\beta^c(\cdot)$ and $\beta^p(\cdot)$ are functions to indicate the base attributes in $\mathbf{B}^c$ and $\mathbf{B}^p$ for an attribute $a$, respectively.

Now, given two seen attributes $a_k$ and $a_l \in \mathbf{A}^s$ in which $a_k = \{\beta^c(a_k), \beta^p(a_k)\}$ and $a_l = \{\beta^c(a_l), \beta^p(a_l)\}$, if $a_k$ and $a_l$ have common ground in either the base attribute of adjectives (i.e. $\beta^c(a_k) = \beta^c(a_l) \in \mathbf{B}^c$) or the one of object parts (i.e. $\beta^p(a_k) = \beta^p(a_l) \in \mathbf{B}^p$) but not both, then we can use the **intersection operation** $\mathbb{I}$ to extract such common base attribute from $a_k$ and $a_l$:

$$\mathbb{I}(a_k, a_l) = \begin{cases} \beta^c(a_k) & \text{if } \beta^c(a_k) = \beta^c(a_l), \ \beta^p(a_k) \neq \beta^p(a_l) \\ \beta^p(a_k) & \text{if } \beta^c(a_k) \neq \beta^c(a_l), \ \beta^p(a_k) = \beta^p(a_l) \end{cases} \tag{4}$$

For instance, $\mathbb{I}$ is able to extract the base attribute "red" from the seen attributes "red wing" and "red breast"; or the base attribute "tail" from the seen attributes "buff tail" and "black tail".

Once we obtain the base attributes via intersection over seen attributes, we further adopt the **union operation** $\mathbb{U}$ to create novel attributes. Given two pairs of seen attributes $\{a_k, a_l\}$ and $\{a_{k'}, a_{l'}\}$ in which $\beta^c(a_k) = \mathbb{I}(a_k, a_l)$ and $\beta^p(a_{k'}) = \mathbb{I}(a_{k'}, a_{l'})$, i.e. $\{a_k, a_l\}$ share the same base attribute

of adjective while $\{a_{k'}, a_{l'}\}$ share the same base attribute of object part, a novel attribute $\tilde{a}$ can be synthesized by combining $\beta^c(a_k)$ and $\beta^p(a_{k'})$, i.e. $\tilde{a} = \mathbb{U}(\beta^c(a_k), \beta^p(a_{k'}))$. In particular, if such combination of base attributes has been seen in $\mathbf{A}^s$, i.e. there exists an attribute $a \in \mathbf{A}^s$ where $\beta^c(a) = \beta^c(\tilde{a})$ and $\beta^p(a) = \beta^p(\tilde{a})$, we say the seen attribute $a$ is **reconstructed** by $\tilde{a}$. Otherwise, if none of the seen attributes has the identical combination as our synthesized $\tilde{a}$, we denote $\tilde{a}$ a **novel attribute** and $\tilde{a} \in \mathbf{A}^u$. In summary, extracting base attributes from seen attributes via intersection, followed by combining the base attributes into novel attributes via union, holistically forms our **decompose-and-reassemble** procedure to synthesize the novel attributes.

In practice, our implementation of the intersection function $\mathbb{I}$ (please refer to its illustration provided in Appendix. A.2) is built based on the encoder architecture of vision transformer [10] (ViT) , in which its input is the embeddings of the seen attributes, i.e. the model takes $m_k^s$ and $m_l^s$ from $M^s$ as input when performing $\mathbb{I}(a_k, a_l)$, where $a_k, a_l \in \mathbf{A}^s$. To be detailed, there are several modifications in our model for intersection $\mathbb{I}$ with respect to the original ViT: (1) We remove the position embedding in order to fulfil the commutative property of intersection, i.e. $\mathbb{I}(a_k, a_l) = \mathbb{I}(a_l, a_k)$; (2) We attach a learnable token named "intersection head" to the input sequence of transformer, which is similar to the extra class embedding in ViT. The corresponding output of this intersection head after going through the transformer encoder represents the embedding of the resultant base attribute, where we apply the element-wise absolute-value operation on it to make it a non-negative vector (being analogous to what we did for the seen attributes); (3) There is only a single self-attention block in the transformer. Please note that, the embedding of a base attribute is also a $C$-dimensional vector. Regarding our union function $\mathbb{U}$, we simply adopt the average operation for its implementation, that is: Given two base attributes $b^c \in \mathbf{B}^c$ and $b^p \in \mathbf{B}^p$, we obtain the embedding $\tilde{m}$ of the synthesized attribute $\tilde{a} = \mathbb{U}(b^c, b^p)$ by averaging the embeddings of $b^c$ and $b^p$. Specifically, such $C$-dimensional embedding $\tilde{m}$ is also defined upon the image feature and acts as the detector for the synthesized attribute $\tilde{a}$.

The training of our decompose-and-reassemble procedure for synthesizing novel attributes is simply based on the reconstruction loss of the seen attributes $\mathcal{L}_{rec}$. Given a synthesized attribute $\tilde{a}$, if there exists a seen attribute $a_k \in \mathbf{A}^s$ with having $\beta^c(a_k) = \beta^c(\tilde{a})$ and $\beta^p(a_k) = \beta^p(\tilde{a})$, the embedding $\tilde{m}$ of $\tilde{a}$ and the embedding $m_k^s$ of $a_k$ are expected to be identical to each other, $\mathcal{L}_{rec}$ is thus defined as:

$$\mathcal{L}_{rec} = \|m_k^s - \tilde{m}\| \tag{5}$$

Note that, as our union function $\mathbb{U}$ has no trainable parameters (since it is simply an average operation), the gradient of $\mathcal{L}_{rec}$ is propagated to focus on learning the parameters of our transformer for the intersection function $\mathbb{I}$. In other words, we expect that the transformer is so powerful to be capable of extracting the base attributes where their averages are informative enough to act as the detectors for the synthesized attributes. Note that in order to fully leverage the seen attributes for training our decompose-and-reassemble procedure, we have a particular training scheme where its algorithm and the implementation details are shown in the Appendix. A.1.

## 4   Experimental Results

**Dataset.** Our experiments are mainly conducted on the Caltech-UCSD Birds-200-2011 dataset [35] (usually abbreviated as CUB) for zero-shot classification. CUB dataset collects 11,788 images of 200 bird categories, where each image is annotated with 312 attributes. We select 32 attributes as our seen attributes $\mathbf{A}^s$, which can be decomposed into 15 base attributes of adjective $\mathbf{B}^c$ and 16 base attributes of object part $\mathbf{B}^p$, and we use these base attributes to synthesize 207 novel attributes $\mathbf{A}^u$. We follow the setting proposed by [38] for the task of generalized zero-shot learning (GZSL) to split the CUB dataset, where such training and testing sets are used to train and evaluate our proposed scenario of ZSL on attributes, respectively. Furthermore, we base on [13] to create a synthetic dataset, named $\alpha$-CLEVR, for providing extensive analysis on the robustness of our proposed ZSLA against the noisy seen attributes (which are related to an issue hidden behind CUB dataset). $\alpha$-CLEVR contains 24 attributes (i.e. types of toy bricks) which are the combinations of 8 colors (i.e., base attributes of adjective $\mathbf{B}^c$) and 3 shapes (i.e., base attributes of object part $\mathbf{B}^p$). Among them, 16 attributes, which can be decomposed into the 11 base attributes, are selected as seen attributes $\mathbf{A}^s$ for training our proposed ZSLA, where we are then able to synthesize novel attribute detectors to annotate the images of $\alpha$-CLEVR dataset. Noting that, classes in our $\alpha$-CLEVR dataset are defined by the specific combinations of toy bricks (where toy bricks with different color-shape combinations are treated as different attributes) For performing the GZSL task on $\alpha$-CLEVR and evaluate the annotation quality

Table 1: Evaluation of synthesized novel/unseen attributes on attribute classification (mAUROC), retrieval (mAP@50), and localization (mLA). $N^s$ is the number of seen attributes.

| | mAUROC | | | mAP@50 | | | mLA | | |
|---|---|---|---|---|---|---|---|---|---|
| $N^s$ | 32 | 64 | 96 | 32 | 64 | 96 | 32 | 64 | 96 |
| **A-LAGO** | 0.600 | 0.612 | 0.627 | 0.173 | 0.180 | 0.222 | 0.782 | 0.787 | 0.795 |
| **A-ESZSL** | 0.626 | 0.614 | 0.632 | 0.223 | 0.200 | 0.234 | 0.756 | 0.769 | 0.756 |
| **Our ZSLA** | **0.689** | **0.704** | **0.717** | **0.320** | **0.327** | **0.329** | **0.846** | **0.860** | **0.867** |

of our synthesized attribute detectors, we create 160 classes in $\alpha$-CLEVR, where each class has 30 images; 80 classes are set as seen data, and the other 80 are set as unseen data. For GZSL inference, testing images from both seen and unseen classes are used.

**Baselines.** As our task of ZSL on attributes for dataset annotation is novel, there is no prior work that we can directly make a comparison with. However, as the task of ZSL on attribute has a hierarchy between attributes and base attributes, we adapt two representative methods of zero-shot classification (which explicitly have the class–attribute hierarchy behind their formulation) to be our baselines, by using the analogy between two hierarchies (i.e. our attribute–base attribute versus their class–attribute). These two baselines are ESZSL [32] and LAGO singleton [3] (note that both of them realize classification with the help of attribute prediction), in which we particularly rename their adaptions to our scenario of ZSL on attributes as **A**-**ESZSL** and **A**-**LAGO** respectively for avoiding confusion. Details of implementing baselines are provided in Appendix C.3.

Note that, in the following experiments, both the baselines and ZSLA use the additional ground-truth of attribute locations (i.e. knowing where an attribute appears on the image) provided by CUB to train the seen attribute detectors, unless stated otherwise.

**Limitation and discussion.** Our proposed method (ZSLA) tackles zero-shot learning for unseen attributes and enables the efficient attribute annotations when constructing the new datasets (later shown in Sec 4.2) as long as the attribute format can be re-factored (e.g. blue wing) for assembling novel attributes via the decompose-and-reassemble approach. Although such requirement on the attribute format restricts the direct application of our ZSLA on some ZSL datasets (e.g. AWA2 and SUN datasets), the particular attribute format adopted in our ZSLA actually better fits the human's intuition for describing the discriminative attributes for the "fine-grained" classes. Hence, we would not see this format requirement on attributes as the limitation of our proposed method but the restriction of the mainstream ZSL datasets for not having the fine-grained attributes.

## 4.1 Evaluation of Unseen Attributes

We design three schemes to evaluate the quality of the synthesized novel attribute detectors learnt by ZSLA: (1) **Attribute Classification.** Based on ground-truth attribute annotation of the test images (note that each image typically has multiple attributes), we measure the performance of our synthesized attribute detectors on recognizing their corresponding attributes in the test images. We adopt the area under receiver operating characteristic (AUROC) as our metric for the classification accuracy of each attribute, and we report the average over AUROCs (denoted as mAUROC) of all synthesized attribute detectors; (2) **Attribute Retrieval.** We rank the test images according to their image-wise responses as to a given attribute detector, to simulate the application scenario of retrieving the images which are most likely to own the target attribute from an image set. Note that the image-wise response is computed by max-pooling over the responses of patch-wise image features with respect to the attribute detector. For each attribute detector we compute the average precision (AP) of its top 50 retrieved images, and report the average AP (denoted as mAP@50) of all detectors as the metric; (3) **Attribute Localization.** As in CUB the ground-truth locations that an attribute appears on the test images are available, we introduce the localization accuracy (LA) to measure how well the location having the highest response to an attribute detector matches with the ground-truth ones (counted as correct if they are located on the same or neighboring patches). We average over the LA of each attribute as the metric (denoted as mLA).

Table 1 summarizes the performance in terms of mAUROC, mAP@50, and mLA obtained by baselines and ZSLA, with the number $N^s$ of seen attributes $\mathbf{A}^s$ set as $\{32, 64, 96\}$. It is clear to see that ZSLA provides superior performance in comparison to the baselines on all the settings of $N^s$

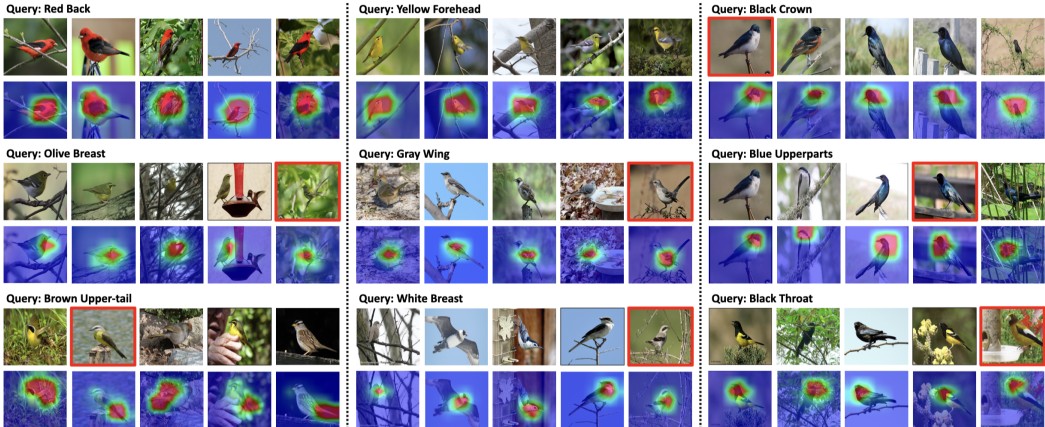

Figure 3: Examples of attribute retrieval and localization. Each set shows the top-5 retrieved images and their response maps for a synthesized novel attribute, where the images marked with red borders are the false positives according to CUB ground-truth.

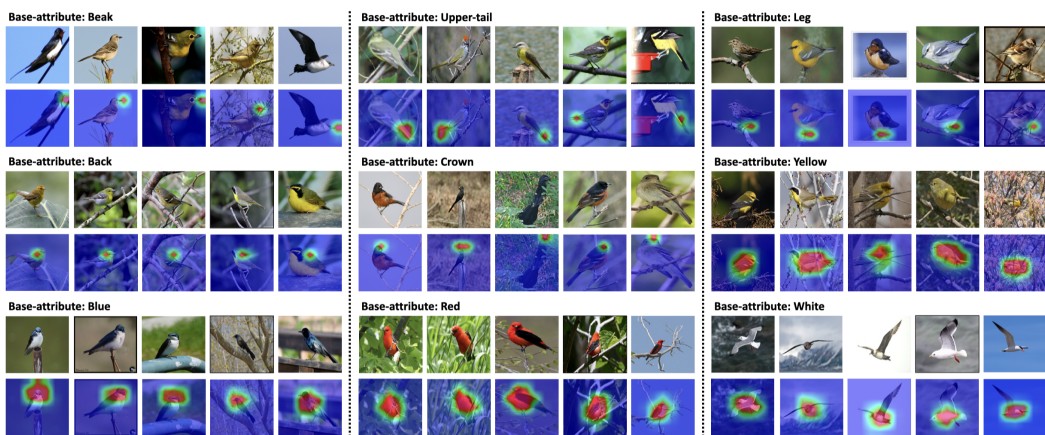

Figure 4: Examples of showing the retrieval and localization ability of base attributes. Each set shows the top-5 retrieved images and their corresponding response map for a base attribute representation (extracted by applying our intersection operation on seen attributes detectors).

and evaluation schemes, particularly the localization accuracy. Moreover, by using merely 32 seen attributes to perform the synthesis of novel attribute detectors, ZSLA can achieve comparable results with the baselines of using 64 or 96 seen attributes. Qualitative examples for showing the results of attribute retrieval and attribute localization for the novel attributes synthesized by ZSLA are provided in Figure 3. Besides these quantitative and qualitative results demonstrating the efficacy of ZSLA on novel attributes, we also provide some qualitative examples in Figure 4 to showcase the localization and retrieval ability of our base attribute representations extracted from the seen attribute detectors.

## 4.2 Automatic Annotations for Learning Generalized Zero-Shot Image Classification

To further access the quality of our synthesized attribute detectors, we adopt the 32 seen attribute detectors and the 207 novel attribute detectors (i.e. $N^s$=32, $N^u$=207) learned by ZSLA to *re-annotate the attribute labels for the whole CUB dataset* to simulate the labeling process during constructing a new dataset, and name the resultant new dataset "$\delta$-CUB". Then we adopt $\delta$-CUB to train and evaluate four representative GZSL algorithms, i.e. ALE [1], ESZSL [32], CADAVAE [33], and TFVAEGAN [26] using the settings proposed by [38] (i.e. for $\delta$-CUB and CUB, training with samples from the 150 seen classes, then evaluating the performance on all 200 classes including the 50 unseen ones). Note that, the class-attribute matrix, which shows the composition of attributes for each class and is needed for GZSL (i.e. the semantic information of classes), is computed by the statistics in

Table 2: Experiments results of training and evaluating four representative GZSL methods (i.e. CADAVAE, TFVAEGAN, ALE, ESZSL) on the datasets built upon different sources of attribute annotation (e.g. manual annotation given by original CUB dataset, and re-annotation provided by ZSLA or baselines). As for the columns, **S** and **U** represent the accuracy on seen and unseen classes respectively, while **H** represents the harmonic mean of **S** and **U**. The highest scores are marked in bold red, while the second-highest ones are marked in bold blue. Particularly, we encourage the readers to observe the relative improvement/gain (in terms of harmonic mean) produced by various approaches/settings with respect to the results obtained by using 32 manually-labelled seen attributes for GZSL (i.e. the results on the blue-shaded row for CUB dataset), where the superior gain contributed by our propose ZSLA well demonstrates its practical value of automatically producing high-quality annotations on the novel attributes.

| | CADAVAE [33] | | | TFVAEGAN [26] | | | ALE [1] | | | ESZSL [32] | | |
|---|---|---|---|---|---|---|---|---|---|---|---|---|
| | S | U | H | S | U | H | S | U | H | S | U | H |
| Manual ($N^s$=32 for CUB) | 42.9 | 27.3 | 33.4 | 45.5 | 31.2 | 37.1 | 26.4 | 9.2 | 13.7 | 29.8 | 10.8 | 15.9 |
| Manual ($N^s$=312 for CUB) | **53.5** | 51.6 | 52.4 | **64.7** | 52.8 | **58.1** | **62.8** | 23.7 | 34.4 | **63.8** | 12.6 | 21.0 |
| A-LAGO | 45.4 | **55.4** | 49.9 | 57.4 | **53.0** | 55.1 | 51.8 | **27.2** | **35.6** | 49.7 | **17.1** | **25.4** |
| A-ESZSL | 41.5 | 48.7 | 44.8 | 56.0 | 48.5 | 52.0 | 49.7 | 17.1 | 25.4 | 61.3 | 9.2 | 16.0 |
| Our ZSLA ($N^s$=32, $N^u$=207 for δ-CUB) | **50.3** | **56.4** | **53.2** | **59.0** | **55.9** | **57.4** | **52.4** | **27.5** | **36.1** | **65.1** | **16.4** | **26.2** |

δ-CUB. Similarly, we also use **A**-**ESZSL** and **A**-**LAGO** baselines to re-annotate CUB dataset and perform GZSL under the same aforementioned setting. The results related to ZSLA and baselines are summarized in the row shaded by the orange color of Table 2. Moreover, we additionally experiment on training the four GZSL algorithms by using only 32 attributes or using all 312 attributes obtained from the original CUB dataset as the semantic information, where their results are summarized in the rows shaded by the blue and green color of Table 2, respectively.

From the results, we observe that using δ-CUB for training, where our ZSLA automatically annotates all the attribute labels, can largely benefit the performance of GZSL algorithms. By treating the harmonic mean over the accuracy numbers on both seen and unseen categories as the metric for GZSL, δ-CUB is superior to those datasets annotated by baselines or even the one using manual annotations. Specifically, the gain obtained by using our δ-CUB with respect to the setting of using 32 manually-labeled attributes (i.e., the blue-shaded row of Table 2) demonstrates the practical value of our proposed problem scenario of ZSL on attributes: Without additional cost for collecting annotation, we provide more attribute labels via synthesizing novel attribute detectors from the seen ones, and thus different categories can be better distinguished by more fine-grained/detailed attribute-based representations. Moreover, regarding the results that our automatic re-annotation leads to better performance than the manual one (i.e., the green-shaded row of Table 2), we believe that this is mainly due to the biased semantic information caused by noisy labels stemming from the inconsistency between different human annotators when building CUB dataset. In comparison, our attribute detectors can produce consistent attribute annotations as we use the same set of attribute detectors for labeling all images; it eventually contributes to a more suitable semantic for learning zero-shot classification. We provide more discussions on such issues in Appendix C.1

### 4.3 Robustness Against Noisy Attribute Labels

Due to the preference bias among different annotators mentioned in Section 4.2, it is hard to obtain perfect seen attribute labels for training. Thus, it is interesting to discuss the effect of the noisy level of seen attribute labels (used for training) on the final annotation quality produced by different auto-annotation methods. Thus, we conduct the controlled experiments using α-CLEVR to understand the effect of the noisy labels. In detail, we adjust the amount of noisy labels for the analysis. To measure the performance drop caused by noisy seen attribute labels, we define the **wrong attribute label rate** (abbreviated as **WALR**) to represent the noisy level of attribute labels. For example, when WALR is set to 0.3, any toy brick in the training images has a 30% chance of inaccurately annotating (e.g., a blue cube is annotated as a red sphere).

As the mAUROC curves shown in Figure 5(a), we can observe that: (1) ZSLA outperforms the baselines in attribution classification, no matter how noisy the training data is; (2) the performance

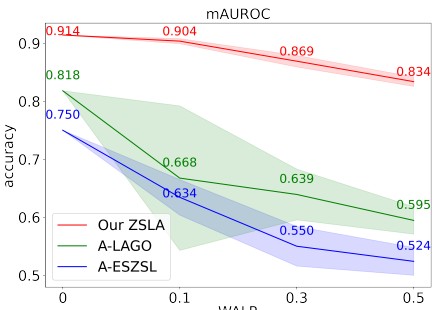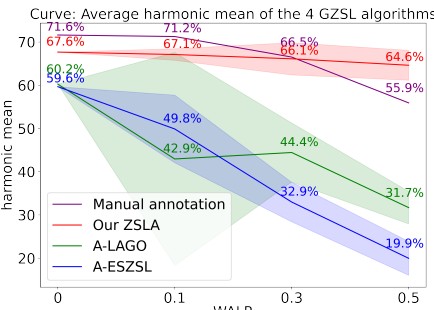

Figure 5: (a) Left: Evaluation (in terms of attribute classification, with mAUROC as the metric) on the robustness against noisy attribute labels for various methods which learn to synthesize the novel attributes. (b) Right: Evaluation on the quality of automatic re-annotation produced by different methods, where the performance is based on the average harmonic mean of four GZSL algorithms using the re-annotated attributes (cf. the last paragraph of Sec. 4.3 for details). Noting that each curve reports the average performance over 5 runs of different noisy label sets (as the label noise is randomly injected), while the shaded bands around each curve represent the 95% confidence interval.

drop of ZSLA with respect to WALR is much smaller than that of the baselines; (3) baselines have a larger variance than ours, i.e., they are more sensitive to different combinations of the noisy labels even the noise level is the same. The observations prove the robustness of ZSLA against the noisy labels of training attributes. We also show in Figure 13 in Appendix that mAP@50 and mLA (for attribute retrieval and location) have a similar trend as mAUROC.

Moreover, similar to the experimental setting as Section 4.2, we use the novel attribute detectors, synthesized by different methods under various WALR settings, to automatically re-annotate the dataset. The resultant dataset is used for learning four GZSL algorithms (i.e., CADAVAE, TFVAEGAN, ALE, and ESZSL). The average of their harmonic means is reported in Figure 5(b). We can observe the superior quality in terms of automatic re-annotation produced by our ZSLA (i.e., the red curve) compared to the other baselines (i.e., the blue and green curves for A-ESZSL A-LAGO, respectively) under all WALR settings. Specifically, we also simulate the situation where humans annotate all attribute labels for the dataset while maintaining the corresponding WALRs (i.e., the purple curve). It leads to a similar observation as we find in the CUB dataset. Once WALR is high (i.e., quite noisy labeling), the performance of GZSL algorithms trained with the semantic information provided by our ZSLA (i.e., the red curve) becomes superior to the one trained with the noisy manual labels.

## 5    Conclusion

This paper proposes a new method of developing zero-shot learning on novel attributes to reduce the attribute annotation cost for constructing a zero-shot classification dataset. By leveraging the trained detectors of seen attributes, our model learns to decompose them into base attributes to further synthesize novel unseen attributes by reassembling pairs of base attributes. Experimental results show that our method is able to exploit the information embedded in the seen attributes to generate high-quality unseen attributes, validated by various evaluation schemes for attribute classification, retrieval, and localization. We also demonstrate that the semantic information based on our automatic re-annotation is beneficial for the GZSL task.

**Acknowledgement.** This project is supported by NSTC (National Science and Technology Council, Taiwan) 111-2636-E-A49-003, 111-2628-EA49-018-MY4, 110-2221-E-A49-066-MY3 and 109-2221-E-009-112-MY3. We are grateful to the National Center for High-performance Computing for computer time and facilities.

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
