# OpenReview forum: "Make an Omelette with Breaking Eggs: Zero-Shot Learning for Novel Attribute Synthesis"
_NeurIPS.cc/2022/Conference — NeurIPS 2022 Accept_

### Official Review · Reviewer_Vc5a · 2022-07-11

**Rating:** 6
**Confidence:** 4
**Soundness:** 3 good
**Presentation:** 2 fair
**Contribution:** 2 fair

**Summary:**

The paper proposes an approach for estimating new detectors of visual attribute conjunctions on images by unmixing its parameters and recombining them. The attribute detector parameter is a prototype in deep feature space that localizes the attributes by cosine similarity. The recombination is produced by averaging the detector parameters. The new attribute synthesis can be used to solve Zero-Shot Classification (ZSC) problems when only few attribute annotations are available in the learning dataset by augmenting the annotations with new synthesized attribute detectors. The approach is compared to 4 other ZSC algorithms on the CUB dataset.


**Questions:**

- When reading in the supplementary (C.2) the way the unseen categories are described in $\delta$-CUB dataset from a series of images, it seems to me that what is actually evaluated is more a “few-shot”  than a “zero-shot” learning scheme since images are needed to compute the description: isn’t there a bias when comparing to “pure” GZSL approaches which have to deal with noisy attribute description of unseen categories?
- I didn’t understand what is the input of the network computing the intersection : is it just an attribute label? If so, how is it encoded? Or is it the detector parameter $m$?
- Exploiting a VIT architecture seems to me oversized given the complexity of the dataset. Why such a model?
- The question of building atomic attributes from their conjunctions reminds me of the classical question of blind source separation in signal processing: is there any similarity to that problem?


**Limitations:**

Non applicable

**Strengths And Weaknesses:**

Strengths
- Addressing knowledge transfer at the level of attribute description for classifier design is an interesting idea.
- Rather detailed experiments (distributed b/w main paper and supplementary material) although on a single small dataset.

Weaknesses
- The approach is validated on a single dataset (CUB), limiting its potential usefulness.
- A lot of material is provided in the main paper and supplementary material (17 pages!) but in a confusing organization. For instance, the main algorithm (“intersection”) is not fully described in the main paper, and we need to search for the information in the supplementary material. Wordy general discussions could be replaced fruitfully by more specific technical details.
- The Zero-shot character of the proposed approach seems to me over-stated, and thus misleading: the new attribute synthesis looks like an unmixing/mixing problem, and the unseen category description with the new attributes in fact requires images, which breaks the “zero-shot” principle, as far as I understand it.

---

> ### Author Response · Authors · 2022-08-02
> **Response to Reviewer Vc5a - Part 3**
>
> ### Q4. [The input and architecture design of intersection model ]
> >*“I didn’t understand what is the input of the network computing the intersection: is it just an attribute label? If so, how is it encoded? Or is it the detector parameter m?”*
>
> As described in line 198-199 of our main manuscript, the input of the intersection model is the weight vectors (i.e. embeddings) of two trained seen attribute detectors.
>
> >*“Exploiting a VIT architecture seems to me oversized given the complexity of the dataset. Why such a model?”*
>
> The VIT-based architecture we adopt for our intersection model is shown in Figure 1 of Appendix, where only a single self-attention block is used (as described in line 42-43 of Appendix A.3, and we also include the corresponding description in the revised main manuscript) thus our intersection model is actually light-weight and not oversized (this implementation detail can be also seen in the source code provided in supplementary file: /code/networks/logic_network.py). The main reason behind our design choice of adopting VIT-based architecture is due to the fact that the transformer framework without position embedding (as described in line 200-206 in our main manuscript) nicely satisfies the commutative property needed for our intersection operation.
>
> ### Q5. [Relation to Blind Source Separation (BSS)]
>
> >*“The question of building atomic attributes from their conjunctions reminds me of the classical question of blind source separation in signal processing: is there any similarity to that problem?”*
>
> We thank the reviewer for bringing up such an insightful question for discussion. Though our ZSLA framework composed of decompose-and-reassemble procedure seems to be similar to blind source separation at the first glance, there actually exists a significant distinction which clearly differentiates our ZLSA from the blind source separation in signal processing: Our intersection operation to perform decomposition on seen attributes is actually non-blind, in which it works by the guidance of logical and semantic constraints (i.e. two input attributes should have a common ground in one of the base attributes but not both), while there is no such constraint in the blind source separation (which instead typically adopts independent assumption or mutual information in its modeling) and it could be non-trivial to include logical/semantic constraints into the framework of blind source separation. We will add this discussion in Appendix A.5 in the revised version.

---

> ### Author Response · Authors · 2022-08-02
> **Response to Reviewer Vc5a - Part 2**
>
> ### Q3. [Pure zero-shot learning]
> >*“The Zero-shot character of the proposed approach seems to me over-stated, and thus misleading: the new attribute synthesis looks like an unmixing/mixing problem, and the unseen category description with the new attributes in fact requires images, which breaks the “zero-shot” principle, as far as I understand it.”*
>
> As mentioned, the concept of our proposed zero-shot learning for attributes scenario is related to the level of **“attributes ↔ base-attributes”** in the hierarchy. Our ZSLA achieves the goal by decomposing the base attributes out from the seen attribute (where these base attributes are hidden in the seen attribute thus being implicitly observed) and then reassembling the base attributes to compose the unseen attributes (i.e. having zero-shot for these composed unseen attributes), in which such scenario is actually analogous to the zero-shot classification (where the attributes are implicitly observed in the seen classes and further used to compose or define the unseen classes) thus following the general definition of “zero-shot” learning (i.e. the auxiliary semantic information is implicitly observed during the training and is used to define the unseen targets).
>
> In brief, the comment from the reviewer “the unseen category description with the new attributes in fact requires images” is basically wrong, due to the facts that: 1) our ZSLA is in the level of **“attributes ↔ base-attributes”** and it learns to synthesize novel attributes without any dependency on the category information; 2) the unseen category description can be tackled by any existing zero-shot classification algorithms (e.g. CADAVAE [22],  TFVAEGAN [19], ALE [1], and ESZSL [21] used by our experiments in Section 4.2) without requiring to see the corresponding images.
>
> >*“...When reading in the supplementary (C.2) the way the unseen categories are described in δ-CUB dataset from a series of images, it seems to me that what is actually evaluated is more a “few-shot” than a “zero-shot” learning scheme since images are needed to compute the description…”*
>
> **Here we would like to have a clarification first: the main purpose of the experiments based on $\delta$-CUB dataset is to evaluate the quality of the automatic attribute annotations produced by our synthesized attribute detectors, where these attribute detectors learned by our ZSLA are independent of the categories/classes.**
>
> Moreover, our ZSLA is shown to contribute to construct the zero-shot classification dataset (e.g. $\delta$-CUB) with little cost (as demonstrated in the experiments of Section 4.2 of our main manuscript, based on merely 32 seen attributes, we can synthesize another 207 novel attribute detectors, where the resultant automatic attribute annotations in $\delta$-CUB can largely benefit the performance of GZSL algorithms). The overall procedure to build up the $\delta$-CUB dataset by our ZSLA is summarized as follows:
>
> 1. The human annotators annotate several types of attribute labels for the CUB dataset.
> 2. The annotated attribute labels are utilized for training a set of seen attribute detectors.
> 3. The weights of the trained seen attribute detectors are used for training our intersection and union models.
> 4. We synthesize the weights of the unseen/novel attribute detectors by the trained intersection and union models.
> 5. The synthesized unseen attribute detectors provide annotations for images in the CUB dataset, where the re-annotated images all together construct the resultant $\delta$-CUB dataset.
>
> During the procedure described above, our ZSLA acts as an annotator (just like what human annotators do). No matter in the training phase or in the inference phase of ZSLA, the ZSLA model never obtains any example/labeled images showing how an unseen attribute links to its corresponding visual concept, instead the ZSLA model directly learns to synthesize the unseen attribute detectors via set operations (i.e. intersection and union), which exactly fulfills the basic idea of “zero-shot” learning for the attribute annotations.
>
> >*“...isn’t there a bias when comparing to “pure” GZSL approaches which have to deal with noisy attribute description of unseen categories?”*
>
> **We again clarify that our ZSLA simply acts as an annotator (just like what human annotators do) during constructing the dataset**, and the learning of GZSL algorithms upon the constructed zero-shot classification dataset is totally independent of our ZSLA. Since the relation between CUB and $\delta$-CUB (as described in Section 4 of our main manuscript) is akin to “the same dataset but annotated by different groups of annotators”, the performance of GZSL algorithms thus reflects the quality of the attribute annotations provided in the training dataset.

---

> ### Author Response · Authors · 2022-08-02
> **Response to Reviewer Vc5a - Part 1**
>
> We thank Reviewer Vc5a for reviewing our paper; however, we believe there are some critical misunderstandings in the reviewer's summary and we would like to have a clarification here first:
> As motivated in our introduction section, we assume that there exists a “classes ↔ attributes ↔ base-attributes” hierarchy, in which the typical setting of zero-shot classification (where most existing works of zero-shot learning address) adopts attributes as the auxiliary semantic information to associate classes (e.g., using seen attributes to define the unknown classes) thus being related to the level of “classes ↔ attributes” in the hierarchy, while our proposed zero-shot learning for attributes instead aims to leverage base-attributes as the auxiliary semantic information to link across attributes (e.g., using seen base-attributes to define the unknown attributes) thus being related to the level of “attributes ↔ base-attributes” in the hierarchy. In brief,
> 1. Our ZSLA tackles the “zero-shot learning for attributes” problem (which focuses on synthesizing unseen attributes) instead of “zero-shot learning for categories/classes (also termed as zero-shot classification, or abbreviated as ZSC by the reviewer)” problem. In particular, the attribute detectors produced by our ZSLA can be used to **automatically** annotate the attributes during constructing the training dataset for zero-shot classification (as obtaining the auxiliary semantic information for zero-shot classification requires images being annotated by attributes), thus alleviating the expensive cost of manual annotations.
> 2. As our proposed ZSLA is tackling a different problem (i.e. zero-shot learning for attributes) from zero-shot classification, we do not directly compare our ZSLA with ZSC algorithms. Instead, the baselines (i.e. A-ESZSL and A-LAGO as described in Section 4 of our main manuscript) that we make comparisons with are adapted from their original ZSC settings into our scenario of zero-shot learning for attributes. Moreover,  as described in Section 4.2 of our main manuscript, since our synthesized attribute detectors are able to provide automatic attribute annotations for constructing the ZSC training dataset, we, therefore, adopt four ZSC algorithms (i.e. CADAVAE [22],  TFVAEGAN [19], ALE [1], and ESZSL [21]) to evaluate the quality of attribute annotations.
>
> ### Q1. [The concern on our validation dataset]
> >*"The approach is validated on a single dataset (CUB), limiting its potential usefulness."*
>
> Though our current experiments are mainly conducted on a single dataset (CUB), we view this as a limitation of the mainstream datasets for zero-shot learning, rather than a limitation of our method. Moreover, CUB dataset is for "fine-grained" classification thus typically being treated as the most challenging dataset in zero-shot learning. Furthermore, we would like to emphasize that we also conducted experiments on another dataset, CLEVR, which may be overlooked by the reviewer as some of the results are presented in the Appendix (cf. Appendix B.2 and C.4).
>
> The potential usefulness of our proposed ZSLA is also well demonstrated in our experiments and should not be overlooked: For instance, the re-annotation experiment on CUB (as shown in Table 2 of the main manuscript) indicates that the quality of the attributes annotated by our ZSLA method can be comparable or even superior to that of manual annotations for training various GZSL algorithms. Hence, our proposed ZSLA can be used to perform automatic attribute annotations during constructing the training dataset for zero-shot classification. Moreover, our control experiments on alpha-CLEVR (cf. Figures 6 and 7 of the main manuscript) validate the robust annotation quality of our ZSLA against the noisy/ambiguous (manual) labels of seen attributes in the training data.
>
> ### Q2. [Paper organization issue]
> >*"A lot of material is provided in the main paper and supplementary material (17 pages!) but in a confusing organization. For instance, the main algorithm (“intersection”) is not fully described in the main paper, and we need to search for the information in the supplementary material. Wordy general discussions could be replaced fruitfully by more specific technical details."*
>
> We thank the reviewer for the feedback on the paper organization and would follow reviewer’s suggestion to move more specific technical details into the main paper for our camera ready version.

---

> ### Author Response · Authors · 2022-08-07
> **Kindly inquiring if you had a chance to examine the response**
>
> We thank Reviewer Vc5a for providing constructive and insightful comments. In our response, we believe we have addressed your concerns, which are summarized as follows:
> 1. We clarify the reviewer's critical misunderstanding, in which our zero-shot learning for attribute (ZSLA) is actually different from generalized zero-shot learning (GZSL) or zero-shot learning for classification (ZSC). Assume that there exists a “classes ↔ attributes ↔ base-attributes” hierarchy, then our ZSLA is related to the level of “attributes ↔ base-attributes” while ZSC focuses on the level of “classes ↔ attributes”. Therefore, we do not compare ZSLA with any ZSC algorithms (i.e., CADAVAE [22], TFVAEGAN [19], ALE [1], and ESZSL [21] in Section 4.2 of our main manuscript), but adapt/modify two algorithms from ZSC into the scenario of zero-shot learning for attributes (i.e., A-LAGO and A-ESZSL) as our baselines to make comparison with.
> 1. In “Q1. [The concern on our validation dataset]”, we clarify the concerns on our validation dataset and our usefulness by underlining the experiments carried out on CUB and another dataset, alpha-CLEVR.
> 2. In “Q3. [Pure zero-shot learning]”, we clarify that:
>     (1) *Our ZSLA actually works **during the construction of a zero-shot classification dataset** (where ZSLA acts as an attribute annotator like what human annotators do). Since this process is orthogonal to the class information and ZSLA does not observe any visual examples of unseen attributes, ZSLA well fits the definition of “zero-shot” learning.*
>     (2) *The ZSC algorithms are trained **after the zero-shot classification dataset is constructed**, and the performances are used to evaluate the quality of the attribute annotations.*
> 3. In “Q4 [The input and architecture design of intersection model]”, we answer the questions related to our intersection model architecture and point out the corresponding paragraphs in our main manuscript and Appendix.
> 4. in “Q5 [Relation to Blind Source Separation (BSS)]”, we identify the main difference between BSS and our ZSLA: ZSLA works based on the logical and semantic constraints, which are neither blind nor being aligned with the common assumptions behind BSS (e.g. the independence among sources).
>
> As the deadline of author-review rebuttal is approaching, we would like to know if there is any feedback based on our rebuttal, and is there anything we can do to further clarify the reviewer's concern. Please don't hesitate to let us know!
>
> Paper 4688 authors

---

> > ### Comment · Reviewer_Vc5a · 2022-08-08
> > **Few comments on answers to questions**
> >
> >
> > Dear authors,
> >
> > Thank you for providing detailed answers to my questions. Here are a few comments.
> >
> > [Q1] The demonstration of your approach on a single dataset remains a weakness for me, as noted also by author reviewers. You provided many supplementary experiments to compensate it, which is nice, however it is difficult to see if the good performance  of the new synthesized attribute detectors is due to the specificity of CUB (localized bird parts), or is generic. You have discussed the difference with CZSZL and adapted some of the current approaches: maybe the converse could have been done, adapt your approach to the problems of CZSL using the datasets of this field.
> >
> > [Q2] It would be nice indeed to put the detailed "Intersection" algorithm in the main paper. Another notation that bothered me is to write the intersection using abstract attributes $I(a_k,a_l)$ whereas what is used actually as input are the detector parameters $m_k$.
> >
> > [Q3] Your answer made things clearer. Basically, I have been misled by the use of the "zero-shot" expression related to two different things: the synthesis of unseen attribute detectors, and the use of these attributes in a more classical zero-shot classification, actually exploited in the paper as an evaluation tool of the attribute detector synthesis. I am still not convinced that characterizing your approach as "zero-shot" is well suited to your work.
> >
> > [Q4] This is perhaps the answer that satisfies me the least. The learning problem addressed by attribute synthesis is very small (less than a hundred data samples available) and commutativity can be obtained by much simpler architectures. I didn't find either if the same network is used for the intersection of object and adjectives, which are very different features. I am also quite puzzled by the reassemble principle: a simple average. For me, given the cosine similarity used for attribute detection, it can only work if the image features are already disentangled in some way between objects and adjectives. Hence the necessity to validate the approach on other datasets (cf. Q1).
> >
> > [Q5] There may be more similarities with BSS than you state, and I do not see the "logical/semantic" constraints so difficult to introduce from a formal point of view. I agree that this field has a very different history and vocabulary, but I am OK if you mention it as a further direction of investigation or comparison.
> >
> >
> > As a final remark, I think that, thanks to your answers, I have better understood your work: still It costed me quite a lot of time, meaning that the writing could be improved, reorganized and simplified. I will probably raise my rating after final discussion with the other reviewers.

---

> > > ### Author Response · Authors · 2022-08-08
> > > **Response to "Few comments on answers to questions" [2/2]**
> > >
> > > **[Reply to Q4]**
> > >
> > > Adopting a simplified ViT architecture (without positional encoding and with only a single self-attention block, in which such model is actually lightweight) for the intersection model is a design choice that first comes into our mind for realizing the property of commutativity and it results to perform pretty well. However, we agree that there could be other design choices or network architectures serving the same purpose, hence we are open for any suggestions and will be more than happy to try. Moreover, we use the SAME intersection model for both object parts and adjectives. Even object parts and adjectives are very different features/base-attributes (as commented by the reviewer), our intersection model is trained to be invariant to such difference but concentrates on extracting the common ground (i.e. the base-attribute) between two input attributes. In other words, our intersection model is trained to act as a typical set operation regardless of which type of base-attribute is shared among the input attributes. Furthermore, as our intersection model should function well for any combinations of two input attributes that have a common base-attribute, the learning problem in our proposed method hence is not small (not less than a hundred data samples as commented by the reviewer, since we need to take all the possible combinations of two input attributes into consideration) and that becomes another motivation for us to choose ViT architecture as our base to construct the intersection model due to its powerful learning capacity. Regarding the question of reassembling principle, the main motivations behind our design choice of adopting a simple average to build up the union model are described as follows: as we would like our union model to act as a typical set operation (i.e. the union model should function well for any types of input base attributes) and its output should be effective attribute detectors (e.g. the union between the extracted base-attributes “red” and “wing” would become effective to detect “red wing” on images, and the union between “red” and another extracted base-attribute “beak” will be also effective to detect “red beak” on images), the inputs to our union model (i.e. the base-attributes extracted by the intersection model) should be already representative on their own and disentangled from each other to achieve so. To this end, we decide to make our union model as simple as possible and even have no learnable parameters (where we finally adopt simple average computation for our union model), such that all the training objectives are contributed to learning disentangled and representative/informative base attributes (otherwise, if there are some learnable capacities in the union model, the input base attributes could potentially become less informative as there exists a chance to enhance the combination of base attributes during the union model to synthesize the effective attribute detectors). The experimental results demonstrate that the extracted base attributes under our model design (i.e. having simple average for union model) indeed have disentangled and representative characteristics, as shown in Figure.4 of our main manuscript where the resultant base attributes themselves already have certain retrieval and localization abilities. We will better clarify these motivations and intuitions behind our model design for the union model in our next version (in addition to what we already have now at the end of Sec. 3 in our main manuscript). Finally, the design of our decompose-and-reassemble procedure (i.e. intersection and union models) is also validated on another $\alpha$-CLEVR dataset (cf. Appendix C4).
> > >
> > > ---
> > >
> > > **[Reply to Q5]**
> > >
> > > We thank the reviewer again for bringing up the insightful perspective on the similarity between our work and the blind source separation (BSS). We will have a further investigation on BSS and include the corresponding discussion in the next version as suggested.
> > >
> > > ---
> > >
> > > **[Final remark]**
> > >
> > > **We are also delighted to learn that the reviewer is leaning to increase the review rating score. We are just one post away to answer any follow-up questions you and other reviewers may have!**

---

> > > ### Author Response · Authors · 2022-08-08
> > > **Response to "Few comments on answers to questions" [1/2]**
> > >
> > > First, we thank the reviewer for recognizing our efforts to provide clarification in the rebuttal as well as for his/her consideration to raise the rating on our work. Below we sequentially reply to the remaining concerns in the reviewer's comments.
> > >
> > > ---
> > >
> > > **[Reply to Q1]**
> > >
> > > We believe that the good performance of our synthesized attribute detectors is not simply due to the specificity of CUB as we also testify our proposed method on another dataset, $\alpha$-CLEVR, which has quite different data distribution and properties in comparison to CUB (while the only common ground that CUB and $\alpha$-CLEVR share stems from the fact that they both can be modeled in the form of “classes ↔ attributes ↔ base-attributes” hierarchy). Regarding the reviewer’s suggestion of adapting our proposed method to the problem of CZSL, we find it quite insightful and would be happy to explore more possibilities along such direction.
> > >
> > > ---
> > >
> > > **[Reply to Q2]**
> > >
> > > We thank the reviewer again for his/her constructive suggestion and will refine the paper organization as well as notations accordingly in our next version.
> > >
> > > ---
> > >
> > > **[Reply to Q3]**
> > >
> > > We are glad that our rebuttal does help to resolve the misunderstanding. We use the term “zero-shot” to characterize our approach for learning to synthesize novel attribute detectors due to the fact that no annotated visual examples of these novel attributes are explicitly provided to our approach for learning, in which it is analogous to the setting of “zero-shot” classification where no annotated visual examples of novel classes are observed during model training. We will do our best to clarify such analogy as well as the reason why we adopt the term “zero-shot” in our future version.

---

### Official Review · Reviewer_RG1h · 2022-07-11

**Rating:** 5
**Confidence:** 4
**Soundness:** 2 fair
**Presentation:** 2 fair
**Contribution:** 2 fair

**Summary:**

Motivated by the issue of annotation efforts on attribute labels, which are required for zero-shot learning, this paper develops methods to automatically annotate novel attributes for a dataset. Given seen attributes, the proposed method can detect the unseen attributes via a decompose-and-reassemble manner. Results are demonstrated using the CUB dataset alone. As the task of ZSL on attributes is new, most experiments are specified by the authors.

**Questions:**

As discussed in the limitation of the paper, would the proposed method only work for fine-grained and specific datasets? If it does not work for general, coarse-grained ones like AWA2 and SUN, does it imply this approach has quite limited applications?

**Limitations:**

Yes, the authors have provided a paragraph in the beginning of the experimental section to discuss the limitation of the proposed method.

**Strengths And Weaknesses:**

The problem addressed in this paper is new: automatically annotating unseen attributes for zero-shot learning. While the problem is interesting, is it important to drive the ZSL research? Attributes are among one type of semantic information that relate seen and unseen classes. Besides attributes, word embeddings of labels (via word2vec or BERT) can also be used, which can be learned via unsupervised learning and do not require manual annotation. The experiments do not compare the proposed semantic representations (which requires supervision though) to those unsupervised approaches.
Although the proposed decompose-and reassemble approach is fairly sounded, it is not fully validated by the experiments. The experiments were conducted on only one dataset (CUB), which is only one of the five popular datasets in ZSL benchmarks. CUB is fine-grained and challenging, but others (e.g., AWA2, SUN) have different characteristics and are also important.

---

> ### Author Response · Authors · 2022-08-02
> **Response to Reviewer RG1h**
>
> We thank Reviewer RG1h for reviewing our paper and providing suggestions on additional experiments to enrich our contributions.
>
> ### Q1. [Is it important to drive the ZSL research ?]
>
> Among various types of semantic information, adopting attributes in the study of zero-shot learning is still one of the most popular modeling choices and it has been continuously investigated in many recent research works. Attributes not only well act as the semantics to associate classes/categories but also better fit the intuitive sense for humans to describe things, however, annotating attributes usually requires expensive cost and that becomes its main burden of applications. To this end, our proposed method of zero-shot learning for attributes directly contributes to alleviate such problem, in which our ZLSA is able to offer high quality automatic attribute annotations to construct the zero-shot learning dataset with little cost (as shown in the experiments of Section 4.2 of our main manuscript, based on merely 32 seen attributes, we can synthesize another 207 novel attribute detectors, where the resultant automatic attribute annotations in $\delta$-CUB can largely benefit the performance of GZSL algorithms).
>
> Moreover, we also follow the reviewer’s suggestion to experiment with using word2vec as the class semantics for training the GSZL algorithms (where the word2vec embeddings are provided by [22]). The results are summarized in Appendix D.5 of our revised version (also shown in the table below). From the results, we can observe that the class semantics stemming from the attribute annotations produced by our ZSLA can lead to better performance than those based on word2vec embeddings.
> |                  |       | Word2Vec[22]  | Our ZSLA |
> |:----------------:|:-----:|:--------:|:--------:|
> |                  | **S** | **65.5** |   52.8   |
> | **CADAVAE[22]**  | **U** |   11.3   | **58.1** |
> |                  | **H** |   19.3   | **55.3** |
> |                  |       |          |          |
> |                  | **S** |   45.2   |  **59**  |
> | **TFVAEGAN[16]** | **U** |   28.1   | **55.9** |
> |                  | **H** |   34.7   | **57.4** |
> |                  |       |          |          |
> |                  | **S** | **60.1** |   52.4   |
> |    **ALE[1]**    | **U** |   3.3    | **27.5** |
> |                  | **H** |   6.3    | **36.1** |
> |                  |       |          |          |
> |                  | **S** |   63.5   | **65.1** |
> |  **ESZSL[21]**   | **U** |    1     | **16.4** |
> |                  | **H** |    2     | **26.2** |
>
> ### Q2. [The concern on our validation dataset and useness]
> >*“The experiments were conducted on only one dataset (CUB), which is only one of the five popular datasets in ZSL benchmarks...”*
> >*“As discussed in the limitation of the paper, would the proposed method only work for fine-grained and specific datasets? If it does not work for general, coarse-grained ones like AWA2 and SUN, does it imply this approach has quite limited applications?”*
>
> We understand the reviewers' concern. But we view this challenge as a limitation of the mainstream datasets for zero-shot learning, rather than a limitation of our method. Also, we would like to emphasize that we also conducted experiments on another dataset, CLEVR, which may be overlooked by the reviewer as some of the results are presented in the Appendix (cf. Appendix B.2 and C.4).
>
> Particularly, the re-annotation experiment on CUB, as shown in Table 2 of the main manuscript, indicates that the quality of the attributes annotated by our ZSLA method can be comparable to that of manual annotations for training various GZSL algorithms. Moreover, our control experiments on alpha-CLEVR (cf. Figures 6 and 7 of the main manuscript) validate the robust annotation quality of our ZSLA against the noisy/ambiguous (manual) labels of seen attributes in the training data.
>
> The above properties demonstrate one novel application of our work: providing efficient attribute annotations when constructing the new datasets once the attribute format can be re-factored (e.g. blue wing) for assembling novel attributes via the decompose-and-reassemble approach, in which such format actually better fits the human's intuition for describing the discriminative attributes for the "fine-grained" classes.
>
> Nevertheless, we would like to clarify that the current modeling constraint of ZSLA actually comes from requiring the re-factorable format of attributes (e.g. blue wing) for assembling novel attributes via the decompose-and-reassemble approach, instead of being limited to the fine-grained dataset. We consider relaxing such dependency on the attribute format as a future work of our method.
>
> As AWA2 and SUN datasets do not have such re-factorable attribute format, they are not considered in the experiments for our target scenario of zero-shot learning on attributes (as what we have discussed in the limitation part in Section 4 of our main manuscript and Appendix E.1).

---

> ### Author Response · Authors · 2022-08-07
> **Kindly inquiring if you had a chance to examine the response**
>
> We thank Reviewer RG1h for providing constructive and insightful comments. In our response, we believe we have addressed your concerns, which are summarized as follows:
> 1. We emphasize the contribution of our work by pointing out the importance of using “attributes” as class semantics in zero-shot classification and explain how our ZSLA alleviates the main problem of it (i.e. expensive cost of manual attribute annotations) in “Q1 [Is it important to drive the ZSL research ?]”
> 2. We conduct further experiments of utilizing word2vec as semantic information (to associate across classes) for training the GZSL algorithms, and the results clearly indicate the superiority of adopting the class semantics stemming from the attribute annotations produced by our ZSLA. Please refer to our rebuttal “Q1 [Is it important to drive the ZSL research?]”
> 3. We clarify the concern on our validation dataset and elucidate the unsuitability of AWA2/SUN datasets on the scenario of zero-shot setting for attributes in “Q2 [The concern on our validation dataset and useness]”
>
> As the deadline of author-review rebuttal is approaching, we would like to know if there is any feedback based on our rebuttal, and is there anything we can do to further clarify the reviewer's concern. Please don't hesitate to let us know!
>
> Paper 4688 authors

---

### Official Review · Reviewer_2STe · 2022-07-12

**Rating:** 5
**Confidence:** 4
**Soundness:** 3 good
**Presentation:** 3 good
**Contribution:** 2 fair

**Summary:**

The paper proposed to train the model by decomposing overlapping concepts from labeled annotations and then combining them to create new concepts. After training, the model can be used to provide new annotations which can be further used for supervised training. Experimental analyses are provided to explain the results and show improvement over compared methods.

**Questions:**

1) My biggest concern is regarding the generalizability of the approach. Taking Figure 1 for example, the model is basically requesting the training data and testing data to be limited to a scope of linear combinations of the two concept sets "red, blue, green" and "head, wing, breast". Would the model totally fail when a new concept is introduced, like either "yellow" or "leg" is they have not been seen during training? That is, is the model "simply" decomposing the existing concepts without really generalizing to real unseen concepts. As the analogy in the title, if the model can really "make an omelette with breaking eggs", that would be generalizing to a relevant concept of omelette from egg, not just different combinations of "brown, white, dark" and "shell, yolk, membrane" under the realm of egg.

2) How is the proposed model related to multi-task image-segmentation models? For example, if we have an encoder-decoder model with different prediction heads for colors and parts, that model should also be able to disentangle two types of attributes and predict accordingly for unseen combinations, and it can even interpolate between attributes by, e.g., providing soft predictions. How is the proposed model better in that sense?

3) I'm not too familiar with the CUB dataset, so it might be helpful to provide SOTA results on CUB for direct comparisons.

**Limitations:**

The authors addressed the limitations in Section 4.

**Strengths And Weaknesses:**

The intuition and motivation are well demonstrated and easy to understand. The results on given comparison settings are reasonable.

---

> ### Author Response · Authors · 2022-08-02
> **Response to Reviewer 2STe**
>
> We thank Reviewer 2STe for the efforts and time to review our paper. We address each of your questions below.
>
> ### Q1. [New semantic concept]
>
> We would like to have a clarification here first: As motivated in our introduction section, we assume that there exists a **“classes ↔ attributes ↔ base-attributes”** hierarchy, in which the typical setting of zero-shot classification (where most existing works of zero-shot learning address) adopts attributes as the auxiliary semantic information to associate classes (e.g., using seen attributes to define the unknown classes) thus being related to the level of **“classes ↔ attributes”** in the hierarchy, while our proposed zero-shot learning for attributes instead aims to leverage base-attributes as the auxiliary semantic information to link across attributes (e.g., using seen base-attributes to define the unknown attributes) thus being related to the level of **“attributes ↔ base-attributes”** in the hierarchy.
>
> To be detailed, our proposed zero-shot learning for attributes focuses on decomposing the base attributes out from the seen attributes (where these base attributes are hidden in the seen attributes thus being implicitly observed) and then reassembling the base attributes to compose the unseen attributes (i.e. having zero-shot for these composed unseen attributes), in which such scenario is actually analogous to the zero-shot classification (where the attributes are implicitly observed in the seen classes and further used to compose or define the unseen classes) thus following the general definition of “zero-shot” learning (i.e. the auxiliary semantic information is implicitly observed during the training and is used to define the unseen targets). The question raised by the reviewer is actually referring to the case that even the auxiliary semantic information is unobserved during training (e.g. the unseen base attributes "yellow" or "leg" that the reviewer takes as examples), which is quite different from the typical zero-shot learning problem and there exists no prior zero-shot learning works capable of addressing such challenging case to the best of our knowledge.
>
> Nevertheless, our original intent of the paper title is to make an analogy that we can decompose and reassemble broken eggs into omelettes given that they are made by similar ingredients (but of course not something that is impossible to reassemble such as meats). If the reviewer feels the motivation of using the current title is not clear, we are willing to modify it.
>
> ### Q2. [Relation to multi-task image-segmentation models]
>
> Our proposed method is not directly related to the multi-task image-segmentation models due to the fact that: instead of giving a pixel-wise prediction as segmentation, our attribute detectors learn to focus on the image parts which most likely contain the corresponding attributes and provide the image-level prediction on **whether the attributes exist in an image or not**.
>
> ### Q3. [Predict color and part separately and providing the soft predictions]
>
> We thank the reviewer for bringing up such an insightful question for discussion. The reviewer’s idea of **“predicting color and part separately and providing the soft predictions”** is actually quite similar to the scenario of **“Direct Attribute Prediction (DAP)”** as described in [14] for zero-shot learning on multi-category classification. According to [1], the performance of DAP could suffer since DAP turns to focus on the intermediate tasks (i.e., the ones similar to the color and part detections as suggested by the reviewer) instead of taking care of the main task (i.e., detecting the color-part combination).
>
> In fact, the baseline methods we provided in Section 4 of our main manuscript, i.e., A-LAGO (modified from LAGO-Singleton [3], in which it can be viewed as the relaxed version of DAP, please refer to Appendix C.3) and A-ESZSL (modified from ESZSL [21]), can be implicitly treated as advanced algorithms which utilize different ways to fuse the results of implicit base attribute detectors (e.g., the color and part detectors). We compare our ZSLA with them and demonstrate the superior performance of ZSLA across all the experimental settings with respect to those baselines.
>
> ### Q4. [Camparison to SOTA]
>
> Owing to the novel application scenario of our proposed zero-shot learning for attributes, there exists no prior work which we can directly make comparison with. We hence adapt several representative algorithms of ZSL/GZSL as our baselines (i.e., A-LAGO and A-ESZSL as described in Section 4 of our main manuscript). However, in order to further evince the capability of our proposed ZSLA in this task, the comparisons with the modified state-of-the-art algorithms of compositional zero-shot learning (CZSL, which conceptually has the closest setting to ours) are also provided in Appendix D.3. Our ZSLA outperforms the modified state-of-the-art CZSL algorithms in multiple metrics.

---

> ### Author Response · Authors · 2022-08-07
> **Kindly inquiring if you had a chance to examine the response**
>
> We thank Reviewer 2STe for providing constructive and insightful comments. In our response, we believe we have addressed your concerns, which are summarized as follows:
> 1. We clarify the reviewer’s main concern on the generalization ability of our proposed framework in “Q1 [New semantic concept]”.
> 2. We explain the key distinction between our proposed method and the multi-task segmentation model in “Q2 [Relation to multi-task image-segmentation models]”.
> 3. We summarize the reason why our proposed method is able to outperform the approach of predicting color and part separately and providing the soft predictions (in which such suggested approach is similar to the scenario of Direct Attribute Prediction) in “Q3. [Predict color and part separately and providing the soft predictions]”.
> 4. Owing to the novel application scenario of our proposed zero-shot learning for attributes, there exists no prior work which we can directly make comparison with. However, as mentioned in “Q4. [Comparison to SOTA]”, we have actually provided the quantitative comparison with the baselines respectively adapted from zero-shot classification approaches and the closely-related task (i.e. compositional zero-shot learning), where our ZSLA shows its superiority in multiple metrics.
>
> As the deadline of author-review rebuttal is approaching, we would like to know if there is any feedback based on our rebuttal, and is there anything we can do to further clarify the reviewer's concern. Please don't hesitate to let us know!
>
> Paper 4688 authors

---

### Official Review · Reviewer_2LT8 · 2022-07-24

**Rating:** 6
**Confidence:** 4
**Soundness:** 3 good
**Presentation:** 3 good
**Contribution:** 3 good

**Summary:**

 This paper has proposed a method named zero-shot learning for attributes to deal with a research problem about novel attribute classification and attribute labeling. Specifically, this research problem is tackled by first decomposing the seen attributes into basic attributes, and then recombining these basic attributes into new ones. Experiments are conducted on the CUB and α-CLEVR (a synthetic dataset) datasets for empirical evaluation.

**Questions:**

1. In Sec. 4.2, the four representative GZSL algorithms are a bit archaic, with the earliest being 2013 and the latest being 2020.
2. There is a problem with the case of the title in the references, for example, in Ln. 390, "Clevr" should be "CLEVR".
3. Several NeurIPS conference papers are incorrectly used in the journal paper format, e.g., [6], [9] and [10].

**Limitations:**

In Ln. 257-265, the authors discuss the limitation of the proposed approach that the format of the attribute annotation must be "adjective + object part", e.g., blue wing. However, some ZSL datasets are not labeled with attributes in this form, such as AWA2 and SUN.

**Strengths And Weaknesses:**

- Originality: This paper proposes a novel research problem about novel attribute classification and attributes labeling, which is tackled by zero-shot learning for attributes .

- Quality: In the paper, Sec. 3 details the training of visible attribute detectors with the decomposition-recombination strategy used to synthesize new attribute detectors, providing theoretical support for ZSLA.

- Clarity: The paper is easy to follow and the figures on the paper are straightforwards.

- Significance: The zero-shot problem of attributes is presented and solved, while the proposed method can be used for automatic labeling of attributes.

---

> ### Author Response · Authors · 2022-08-02
> **Response to Reviewer 2LT8**
>
> We thank Reviewer 2LT8 for the positive comments on recognizing our strengths in originality, quality, clarity, and significance. Also, we appreciate the kind reminder for pointing out the problems with our references (e.g., case of the title for the reference paper and wrong citation formats), in which we have fixed them in the revised version. Regarding the concern on the four GZSL algorithms used in Sec. 4.2 being a bit archaic, we believe that these four GZSL algorithms are still representative enough to show promising results as the main purpose here is to demonstrate the quality of the automatic attribute annotations provided by our synthesized attribute detectors. Also, these algorithms are widely adopted, highly cited, and built upon different modeling perspectives (i.e., generative [19, 21] and embedding-based [1, 22] methods).
>
> ### [Extra experiment on another recent GZSL algorithm]
>
> Moreover, we have experimented on another recent GZSL algorithm, CE-GZSL [E1], where the experiments were done by utilizing the official code of [E1] (https://github.com/Hanzy1996/CE-GZSL) with following their default hyper-parameter settings. The experimental setting is the same as described in Section 4.2 of our main manuscript (where different strategies of attribute annotations are applied to build the zero-shot classification dataset, i.e. (re)-annotate the CUB dataset, and GZSL algorithms (e.g. CE-GZSL [E1] in this experiment and CADAVAE [22], TFVAEGAN [19], ALE [1], and ESZSL [21] in Section 4.2 of the main manuscript) are trained on such (re-)annotated dataset to testify the quality of attribute annotations).
>
> In the table below, $N^{s}$ represents the number of attribute types that were annotated manually, while $N^{u}$ represents the number of attribute types that were automatically annotated by the corresponding algorithm. We then train [E1] for solving the GZSL task individually using the semantic information provided by different annotation strategies (i.e. train GZSL on the (re-)annotated dataset by different attribute annotation strategies). The terms S, U, and H respectively indicate the accuracy of the seen classes, the accuracy of the unseen classes, and the harmonic mean of S and U; the numbers in bold represent the best performance. As being observable from the results summarized in the table below, in terms of attribute annotation quality, our ZSLA outperforms the two baselines (i.e., A-LAGO and A-ESZSL) and shows a comparable or even superior result with respect to the fully manual annotated one (i.e., denoted as Manual ($N^{s}=312$)), which reflects a similar tendency as the Table 2 in our main manuscript.
>
> || Manual ($N^{s}=32$) | Manual ($N^{s}=312$) | A-LAGO($N^{s}=32$, $N^{u}=207$) | A-ESZSL($N^{s}=32$, $N^{u}=207$) | Our ZSLA ($N^{s}=32$, $N^{u}=207$)|
> |:-:|:-------------------:|:--------------------:|:------:|:-------:|:-----------------------------------------------:|
> | S |        37.96        |         52.36        |  50.45 |  51.51  |                    **59.84**                    |
> | U |        24.05        |         47.31        |  44.47 |  40.88  |                    **53.28**                    |
> | H |        29.44        |         49.71        |  47.27 |  45.58  |                    **56.37**                    |
>
> [E1] Zongyan Han, Zhenyong Fu, Shuo Chen and Jian Yang. Contrastive Embedding for Generalized Zero-Shot Learning. In *IEEE Conference on Computer Vision and Pattern Recognition (CVPR)*, 2021.

---

> ### Author Response · Authors · 2022-08-07
> **Kindly inquiring if you had a chance to examine the response**
>
> We thank Reviewer 2LT8 for providing constructive and insightful comments. In our response, we believe we have addressed your concerns, which are summarized as follows:
> 1. We fix the problems in the reference part.
> 2. We provide extra new experiments on a recent GZSL algorithm, CE-GZSL, with the same setting as what we have in Section 4.2 of our main manuscript. The experiment results shown in our response share the same conclusion as Table 2 in our main manuscript, which further confirm that the quality of attribute labels automatically annotated by our ZSLA could generally benefit various GZSL algorithms and are superior to manual ones (evaluated by the performance of GZSL algorithms trained on the (re-)annotated zero-shot classification dataset).
>
> As the deadline of author-review rebuttal is approaching, we would like to know if there is any feedback based on our rebuttal, and is there anything we can do to further clarify the reviewer's concern. Please don't hesitate to let us know!
>
> Paper 4688 authors

---

### Meta-Review · Area_Chair_RhUH · 2022-08-26

**Recommendation:** Accept
**Confidence:** Certain

**Metareview:**

This paper has proposed a method named zero-shot learning for attributes to deal with a research problem about novel attribute classification and attribute labeling. The reviewers have many questions in the intial round. After the rebuttal, the authours clarify most unclear points, and some reviewers raise the score. In general, all the reviewers agree with the acceptance of this paper.

**Award:**

No

---

### Decision · Program_Chairs · 2022-09-14

Accept